# Assessing the Impacts of Different Levels of Nano-Selenium on Growth Performance, Serum Metabolites, and Gene Expression in Heat-Stressed Growing Quails

**DOI:** 10.3390/vetsci11060228

**Published:** 2024-05-21

**Authors:** Rania Mahmoud, Basma Salama, Fatmah A. Safhi, Ioan Pet, Elena Pet, Ahmed Ateya

**Affiliations:** 1Department of Nutrition & Clinical Nutrition, Faculty of Veterinary Medicine, Mansoura University, Mansoura 35516, Egypt; ranianutrition@mans.edu.eg; 2Department of Biochemistry and Molecular Biology, Faculty of Veterinary Medicine, Mansoura University, Mansoura 35516, Egypt; dr_basma_salama@mans.edu.eg; 3Department of Biology, College of Science, Princess Nourah bint Abdulrahman University, P.O. Box 84428, Riyadh 11671, Saudi Arabia; faalsafhi@pnu.edu.sa; 4Faculty of Bioengineering of Animal Resources, University of Life Sciences King Mihai I from Timisoara, 300645 Timisoara, Romania; 5Faculty of Management and Rural Tourism, University of Life Sciences King Mihai I from Timisoara, Calea Aradului no.119, 30064 Timisoara, Romania; elenapet@usvt.ro; 6Department of Development of Animal Wealth, Faculty of Veterinary Medicine, Mansoura University, Mansoura 35516, Egypt

**Keywords:** quails, nano-Se, growth, heat stress, oxidative stress, inflammatory markers

## Abstract

**Simple Summary:**

Nanotechnology offers a unique opportunity to incorporate nanoparticles as alternative sources of trace minerals in animal diets. The purpose of this study was to investigate the effects of applying different concentrations of nano-Se on the growth performances, carcass features, serum components, and gene expressions in heat-stressed Japanese quails. The performance of heat-stressed quails was greatly enhanced by the addition of nano-Se, particularly at the level of 0.2 mg/kg of feed, which also repaired the blood oxidative state. The 0.2 nano-Se-exposed group showed a substantial downregulation of the *IL-2*, *IL-4*, *IL-6*, and *IL-8* genes and an upregulation of the *SOD* and *GPX* genes in comparison to the heat stress group, indicating that nano-Se had an ameliorative effect on heat stress. It was determined that modest levels of nano-Se in growing heat-stressed quails yield the best results, and its supplementation can be viewed as a protective management strategy in Japanese quail diets to lessen the adverse effects of heat stress.

**Abstract:**

Nano-minerals are employed to enhance mineral bioavailability thus promoting the growth and well-being of animals. In recent times, nano-selenium (nano-Se) has garnered significant attention within the scientific community owing to its potential advantages in the context of poultry. This study was conducted to explore the impact of using variable levels of nano-Se on the growth performance, carcass characteristics, serum constituents, and gene expression in growing Japanese quails under both thermoneutral and heat stress conditions. A randomized experimental design was used in a 2 × 3 factorial, with 2 environmental conditions (thermoneutral and heat stress) and 3 nano-Se levels (0, 0.2, and 0.5 mg/kg of diet. The findings revealed that heat stress negatively affected the growth and feed utilization of quails; indicated by the poor BWG and FCR. Additionally, oxidative stress was aggravated under heat stress condition; indicated by increased lipids peroxidation and decreased antioxidant enzymes activities. The addition of nano-Se, especially at the level of 0.2 mg/kg of diet, significantly improved the performance of heat stressed quails and restored blood oxidative status. The expression profile of inflammatory and antioxidant markers was modulated by heat stress and/or 0.2 and 0.5 nano-Se in conjunction with environmental temperature in quail groups. In comparison to the control group, the heat stress-exposed quails’ expression profiles of *IL-2, IL-4, IL-6*, and *IL-8* showed a notable up-regulation. Significantly lower levels of the genes for *IL-2, IL-4, IL-6*, and *IL-8* and higher levels of the genes for *SOD* and *GPX* as compared to the heat stress group demonstrated the ameliorative impact of 0.2 nano-Se. The expression profiles of *IL-2, IL-4, IL-6*, and IL-8 are dramatically elevated in quails exposed to 0.5 nano-Se when compared to the control group. *SOD* and *GPX* markers, on the other hand, were markedly down-regulated. It was concluded that nano-Se by low level in heat stressed growing quails provides the greatest performance and its supplementation can be considered as a protective management practice in Japanese quail diets to reduce the negative impact of heat stress.

## 1. Introduction

The Japanese quail has attracted a lot of interest in the poultry industry due to its growing popularity as a source of both meat and eggs. This popularity stems from several key factors, including their rapid growth, which allows them to be ready for human consumption as early as 5 to 6 weeks of age, early sexual maturity, high rate of egg production, and their significantly lower requirements for both feed and space when compared to traditional domestic fowl breeds [1]. To support the healthy growth, development, and overall well-being of quails, it is imperative to provide them with a well-balanced diet that includes a diverse mixture of protein, vitamins, and minerals. These essential minerals play pivotal roles in various physiological functions within the quail’s body. Therefore, when formulating a balanced diet for quails, it is crucial to ensure that they receive adequate amounts of these essential minerals [2].

Selenium (Se) stands out as a vital micronutrient crucial for optimizing poultry performance [3]. It actively contributes to the regulation of various physiological functions, including growth performance, survivability, meat quality, and bolstering antioxidant defenses. Se is incorporated into selenocysteine (SeCys), which is incorporated into selenoenzymes such as glutathione peroxidase (GPx), and this is the reason behind the addition of Se to animals’ feed in order to improve their antioxidant defense [4]. It has been reported that Se bioavailability regulates the activity of GPx enzymes, and supplemental Se generally increases this enzyme activity [5]. Se supplements are usually available as inorganic sodium selenite salt, which is cheap and has low bioavailability, and organic Se in the form of L-selenomethionine (SeMet), which shows higher bioavailability and efficient tissue incorporation, but it is much more expensive to produce. Moreover, it has been reported that SeMet has adverse effects at high doses [6,7,8].

Nanotechnology offers a unique opportunity to incorporate nanoparticles as alternative sources of trace minerals in animal diets, featuring innovative characteristics that are absent in bulk materials or conventional mineral salts [9]. Among these nanoparticles, nano-elemental selenium (nano-Se) has attracted increasing interest in the poultry industry due to its enhanced bioavailability, superior catalytic effectiveness, remarkable absorption capacity, and reduced toxicity compared to traditional selenite sources.

Heat stress is one of the most important environmental stressors challenging poultry production [10]. Heat stress is a significant concern in quail farming, as it can lead to reduced productivity, increased mortality, and health problems among quails. High temperatures can also lead to increased oxidative stress in quails. The production of reactive oxygen species (ROS) in the body can rise under heat stress conditions, causing cellular damage. Antioxidants, such as selenium and vitamins, play a crucial role in neutralizing ROS and protecting quail cells from oxidative harm. Nano-Se supplementation may booster their immune response [11]. It is expensive to cool poultry houses, and methods focus mainly on nutritional modifications. Therefore, there are various nutritional strategies, manipulations, or feeding practices that have been used to aid in reducing and overcoming the negative impacts of heat stress. Nano-Se, with its unique properties, has the potential to offer solutions to counteract the negative impacts of heat stress on growing quails [12].

Nano-Se shows promise as a valuable tool for managing heat stress in growing quails. Its antioxidant, immunomodulatory, and cell-protective properties can help maintain quail health and performance during hot weather conditions [13]. The addition of nano-Se to the poultry diet has been shown to significantly improve feed efficiency, growth performance, and immunological response [14,15]. Moreover, nano-Se supplementation has a notable impact on carcass quality in broilers, without any adverse effects on internal organs [16]. However, there are limited data on nano-Se bioavailability and the mechanism of action in eucaryotic system. Few research studies assumed that nanoelemental Se supplied in a diet could be incorporated into selenoproteins [17,18]. An interesting observation is that elemental selenium showed improved bioavailability and absorption with increased GPx activity, as much as SeMet [15,19]. However, more research data are required to support the knowledge concerning nano-Se’s capability to improve quail performance and its antioxidant ability, especially under the adverse effects of heat stress, with a spotlight on a safe, effective dose for these growing quails. Our driven hypothesis is that varying levels of Nano-Selenium could influence the growth performance, serum metabolites, and gene expression of heat-stressed growing quails. Thus, this study’s main goal was to evaluate the application of different nano-Se levels as feed additives in the diets of heat-stressed quails, on growth performance, carcass traits, tissue selenium content, immune function, serum metabolites, and gene expression in quails.

## 2. Materials and Methods

The animal Care and Use Committee (MU-ACUC) accepted this experimental protocol with the following code number: VM.R.23.07.117. The study was conducted at the Mansoura University Faculty of Veterinary Medicine’s Animal Nutrition and Clinical Nutrition Laboratory.

### 2.1. Green Synthesis of Nano-Se

Nano-Se nanoparticles were synthesized at Nanomaterials Research and Synthesis Unit, AHRI, Egypt, according to the method of Chen et al. (2011), with some modifications [20]. In brief, 4 g of Na_2_SeO_3_ (Loba Chemie, Mumbai, India; M.W = 173.01) and 4 g glucose (ADWIC, Cairo, Egypt) were dissolved in a 100 mL of a mixed solution of ethylene glycol (Merck Schuchardt-Hohenbrunn, Hohenbrunn, Germany) and 100 mL H_2_O. The beakers holding the reactants were sealed and placed in an oven previously set to a temperature of 83 °C for one hour, followed by washing of the produced particles with distilled water several times (minimum of three times). Finally, the particles sizes and distributions were determined using a Zetasizer (Microtrac, Montgomeryville, PA, USA). The nano-Se yielded had an average diameter of 250 nm. The concentration of the final nano-Se suspension was quantified by ICP-MS. 

### 2.2. Quail Husbandry and Housing

A four-week growth trial was conducted at the experimental unit within the Department of Nutrition and Nutritional Deficiency Diseases at the Faculty of Veterinary Medicine, Mansoura University. This experimental facility was equipped with all of the necessary amenities to ensure the well-being of the experimental groups, including appropriate lighting, access to water, a clean and hygienic environment, and adequate ventilation with air suction and fans.

Six equal experimental groups, each including eight duplicates of ten Japanese quails, were created at random from a total of 480 12-day-old Japanese quails. The Japanese quails were housed in plastic wire floor pens measuring 50 × 50 × 40 cm throughout the entire experimental period and lighting was continuously provided for the duration of the experiment, spanning 24 h.

A typical baseline diet was developed in compliance with the guidelines provided by the NRC (1994) [21]. For the experimental diets, nano-Se was incorporated at two different levels as per the method described by Saurai et al. (2006) [6]. The most effective and accurate means of introducing nano-Se into the diets was through wheat bran. Nano-Se was added to the experimental diet at concentrations of either 0.2 or 0.5 mg/kg in a dry form. This was achieved by thoroughly mixing the specified amounts of nano-Se with wheat bran to ensure a uniform distribution [22]. Table 1 lists all of the experimental diets’ ingredients along with the percentages that were determined. To determine the Se content in the diets, an atomic absorption spectrophotometer was employed, yielding the following results: 2.32 mg/kg for the control diet, 3.48 mg/kg for diet 2, and 4.11 mg/kg for diet 3. The Se concentration in diet higher in supplemented groups than in the control group, suggesting dietary nano-Se supplementation increased the Se content in diet.

### 2.3. Experimental Design

A randomized experimental design was used in a 2 × 3 factorial, with 2 environmental conditions (thermoneutral (TN) and heat stress (HS)) and 3 nano-Se levels (0, 0.2, and 0.5 mg/kg of diet). To establish the required thermoneutral condition, the temperature was initially set at 33 °C and was gradually decreased by 2 °C every week. The average ambient temperature during the weeks of the experiment to establish the required heat stress condition was 35 °C using electrical brooders, operating 24 h a day [23].

### 2.4. Measurement of Growth Performance, and Assessment of Carcass Traits Indices

Each quail was weighed separately at the start of the experiment to establish its starting body weight, and at the conclusion, at 40 days of age, to ascertain its body weight increase (BWG). Kilogram feed per kilogram BWG was used to compute the feed conversion rate, or FCR. Following an overnight fast, random 16 quail per replicate were weighed at the conclusion of the experiment before being put to slaughter. 

### 2.5. Sampling

Prior to necropsy, blood samples were drawn into heparinized tubes via the carotid artery and centrifuged for ten minutes at 3000 rpm. Blood plasma samples and lysate from RBCs were kept for analysis at −20 °C. All internal organs were weighed and expressed as a percentage of live body weight following the extraction of feathers, collection of blood, and evisceration. Furthermore, samples of breast muscle were taken from each group, homogenized, dried, and stored at −20 °C for the Se content analysis. Moreover, liver and spleen samples were collected, snap-frozen in liquid nitrogen, and stored at −80 °C until they were used for RNA isolation.

### 2.6. Samples Analyses

#### 2.6.1. Blood Analysis

Following guidelines, erythrocyte lysate was prepared for the analysis of the GPX activity and the reduced glutathione (GSH) level by spectrophotometer using commercially available kits (Biodiagnostic, Giza Governorate, Egypt). Blood plasma samples were used for the measurement of the malondialdehyde (MDA) level and Superoxide dismutase (SOD) levels by spectrophotometer (Photometer 5010, Photometer, BM Co., Hamburg, Germany) and by the enzymatic colorimetric method using commercially available kits (Biodiagnostic, Giza Governorate, Egypt).

#### 2.6.2. Tissue Se Levels Analysis

Samples of muscle (0.5 g) were heated slowly on a plate heater for three hours at 120 °C after being put in a beaker with five milliliters of nitric acid. Following cooling, 10 milliliters of a 5:3 *v*/*v* solution of nitric acid and perchloric acid was added, and to eliminate any remaining organic material, an additional 5 milliliters of nitric acid was added. After heating the mixture to 120 °C until white vapors from the perchloric acid emerged, the concentration of Se was determined using an ICP-MS device [24].

#### 2.6.3. RNA Extraction and Reverse Transcription

The total RNA was isolated from quail spleen and liver with Trizol reagent (Direct-zolTM RNA MiniPrep, catalogue No. R2050), in accordance with the manufacturer’s directives. The quantity and purity were evaluated with a Nanodrop (UV-Vis spectrophotometer Q5000/USA), and the integrity was evaluated using gel electrophoresis. The manufacturing approach (SensiFastTM cDNA synthesis kit, Bioline, catalogue No. Bio-65053) was used to synthesize the cDNA in each sample. The reaction mixture consisted of 20 μL of total RNA, up to 1 μg, and 4 μL of 5× Trans Amp buffer, 1 μL of reverse transcriptase, and 20 μL of DNase-free water. Using a thermal cycler, the reaction mixture was subjected to the following protocol: 10 min of primer annealing at 25 °C, 15 min of reverse transcription at 42 °C, and 5 min of inactivation at 85 °C. The samples were retained in a 4 °C environment.

#### 2.6.4. Quantitative Real-Time PCR

The mRNA levels of inflammatory (*IL-2*, *IL-4*, *IL-6*, and *IL-8*) and antioxidant (*SOD*, and *GPX*) markers were relatively measured using the SYBR Green PCR Master Mix and real-time PCR (2× SensiFastTM SYBR, Bioline, catalog no. Bio-98002). Table 2 shows the primer sequences that were displayed. As an internal control, the housekeeping gene *β-actin* was employed. A total of 0.8 μL of each primer, 5.4 μL of d.d. water, 10 μL of 2× SensiFast SYBR, 3 μL of cDNA, and 10 μL of 2× SensiFast SYBR made up the reaction mix, which was conducted in a complete capacity of 20 μL. The cycling settings for the PCR were as follows: two minutes at 95 °C, forty cycles at 94 °C for denaturation, twenty seconds for annealing at the temperature specified in Table 3, and twenty seconds for an extension at 72 °C. Following the amplification stage, through a melting curve analysis, the PCR product’s specificity was confirmed. The comparative expression of each sample’s gene for *β-actin* in relation to a control gene was determined using the 2^−ΔΔCt^ technique [25].

### 2.7. Statistics

Sample size for assessment of quail performance throughout the experiment was calculated using epi info 7 statestical pachage from the total population using 80% statestical power [26]. 

Data were analyzed by 2-way ANOVA using Graph- Pad Prism v. 7.03 (San Diego, CA), with dietary nano-Se supplementation (0, 0.2, and 0.5 mg/kg of diet) and environmental temperature (TN, HS) as factors. When significant main effects were detected, means were compared using Tukey’s multiple range test. All data are represented as mean ± SEM. 

## 3. Results

### 3.1. Performance Characteristics

Table 3 illustrates the effects of environmental temperature with varying nano-Se levels on final body weight (FBW), body weight gains (BWG), and feed conversion ratio (FCR) of growing quails. When comparing the control group to other groups in the study, several noteworthy observations emerge. The group receiving a basal diet without nano-selenium and subjected to thermo neutral temperature, demonstrated FBW that did not significantly differ from those of the group receiving nano-Se and subjected to thermo neutral temperature. However, the group exposed to heat stress without nano-Se exhibited the lowest BWG and poorer FCR compared to the group exposed to heat stress and supplemented with nano-Se. This suggests that heat stress had a negative impact on these quails’ performance, particularly in terms of BWG and efficient feed utilization. The groups that received 0.2 and 0.5 nano-Se with heat stress exhibited significant improvements in FWB, BWG, and FCR which indicating that nano-Se supplementation, especially at 0.2 levels had a positive impact on quail performance under heat stress conditions.

### 3.2. Carcass Traits

The influence of different nano-Se concentrations and environmental temperature on carcass trait was presented in Table 4. Notably, the different dietary nano-Se levels did not have a significant impact on carcass traits and the relative size of organs in growing Japanese quails. However, high dose of nano-Se (0.5 mg/kg) under heat stress had the lowest numerically dressed carcass weight.

### 3.3. Antioxidant Indices

Blood antioxidant capacity and lipid peroxidation levels were presented in Figure 1. Our results regarding quails that received non-supplemented basal diet with nano-Se (0 mg/kg) revealed that heat stress significantly increased oxidative stress represented by increased MDA level and decreased SOD, GPx, and GSH levels compared to quails raised under thermoneutral condition.

Under heat stress condition, 0.2 mg/kg nano-Se diet supplementation significantly improved quail’s blood oxidative status by decreasing MDA level and increasing SOD, GPx, and GSH levels compared to quails that received non-supplemented basal diet with nano-Se (0mg/kg). Moreover, under thermoneutral condition, 0.2 mg/kg nano-Se supplementation significantly increased antioxidant enzymes levels, SOD, GPx as well as GSH level compared to quails that received non-supplemented basal diet.

On the other hand, it was noticeable that, higher dose of nano-Se supplementation (0.5 mg/kg) showed significant increase in oxidative stress represented by decreased GPx, SOD activities and GSH level with significant increase in MDA level under both thermoneutral and heat stress conditions.

To support our findings so far, we measured Se level in quail’s muscles. As shown in Figure 2, Se levels in muscles significantly increased with increase the concentration of supplemented nano-Se in quail’s diet under both thermoneutral and heat stress conditions. It is worthy noted that under heat stress condition muscle Se significantly decreased in 0.5 mg/kg nano-Se supplemented quails.

### 3.4. Gene Expression Profile of Growth Performance, Immune, and Antioxidant Markers

Quail groups experienced heat stress and/or 0.2 and 0.5 nano-Se alongside environmental temperature could modulate the expression profile of inflammatory and antioxidant markers (Figure 3). Heat stress experienced quails revealed a significant up-regulation in the expression profile of *IL-2, IL-4, IL-6*, and *IL-8* compared to the control group.

The ameliorative effect of 0.2 nano-Se was observed as indicated by significant down-regulation of *IL-2, IL-4, IL-6*, and *IL-8* genes and up-regulation of *SOD* and *GPX* genes compared to heat stress group. Quails exposed to 0.5 nano-Se significantly upsurge the expression profile of *IL-2*, *IL-4, IL-6*, and *IL-8* compared to the control group. Meanwhile, *SOD* and *GPX* markers were significantly downregulated. There was a significant interaction between different concentrations nano-Se and environmental temperature on the expression profile of investigated genes (*p* < 0.01).

## 4. Discussion

Nano-Se supplementation seemed to improve FBW, BWG, and FCR in the heat-stressed quails. The data in the present study indicate that nano-Se supplementation is important for mitigating the adverse effects of heat stress on quail growth and feed conversion. There were no significant differences in the FBWs and BWGs between the control group and the group of quails supplemented with nano-Se without heat stress. This indicates that nano-Se supplementation alone did not have notable impacts on FBW and BWG under normal conditions. In line with our results, Cai et al. (2012) found that supplementing 0.3 to 2.0 mg/kg of Se NPs had no effect on growth performance in broilers, indicating that the impact of nano-Se on growth performance may vary depending on factors like species, dosage, and environmental conditions [27]. However, it is worth noting that a study by Ibrahim et al. (2020) indicated that dietary Che-Se NP (presumably another form of selenium nanoparticles) supplementation significantly improved the BW, BWG, and FCR of broiler chicks compared to the control group [28].

The group exposed to heat stress without nano-Se supplementation had the lowest BWGs and poorer FCRs compared to the control group and other experimental groups, confirming that heat stress negatively impacted quail performance. Our results are in line with the findings of Chen et al. (2015), who also reported a significant decline in FBW, BWG, and FCR under heat stress conditions [29]. Exposure to high temperatures can trigger inflammation in the intestines of poultry. This inflammatory response can lead to a higher risk of a reduced villus/crypt ratio, which is a significant indicator of intestinal health. A lower villus/crypt ratio is associated with the development of chronic enteritis and a decrease in nutrient digestibility, leading to a lower production efficiency in poultry, as the birds are less able to absorb and utilize the nutrients from their feed, resulting in decreased growth and overall performance [30]. Awad et al. (2020) suggested that the growth performance of broilers exposed to heat stress conditions could be attributed to the behavioral, metabolic, and physiological changes that occur in response to a hot environment [31].

On the other hand, heat-stressed quails that received nano-Se at 0.2 and 0.5 mg/kg had a relatively higher FBW and BWG compared to other experimental groups. In addition, the group that was fed a 0.2 mg/kg supplemented diet had the numerically highest BW and BWG. This suggests that nano-Se alone might not significantly affect quail growth, and the positive impact is more pronounced under heat stress conditions. Our results are consistent with a similar trend observed by Safdari-Rostamabad et al. (2017), who found that supplementing broiler diets with nano-Se improved growth performance, the health of internal organs, and the immune response by reducing heat-stress-induced oxidative stress [32]. Moreover, Khazraie and Ghazanfarpoor) 2015) illustrated that weight gain significantly increased in heat-stressed quail chicks fed Che-Se NPs compared to the control [33].

The enhanced FCRs in the groups supplemented with nano-Se may be attributed to the unique properties of nanoform Se, such as its greater surface activity, higher solubility, high cellular uptake, and excellent bioavailability. These properties make nano-Se more effective in terms of utilization [34]. Similarly, El-Deep et al. (2016) reported a significant improvement (*p* < 0.05) in FCRs when broiler diets were supplemented with nano-Se under hot environmental conditions [35]. This alignment with previous research confirms the positive impact of nano-Se supplementation on FCRs, particularly in hot weather conditions.

On the basis of our study, it was evident that altering the dietary levels of nano-Se did not yield significant changes in carcass traits or the relative size of organs. These results are in concurrence with the findings of Khazraie and Ghazanfarpoor (2015), who also reported no substantial effects on carcass traits when nano-Se was incorporated into the diets of chicks under heat stress conditions [33]. However, the administration of a high dose of nano-Se (0.5 mg/kg) under heat stress conditions resulted in the lowest recorded dressed carcass weight. This suggests that the higher nano-Se dose, unlike the lower dose, did not mitigate the effects of heat stress; this could be due to the possibility of retaining more water in the body of quails subjected to this high dose of nano-Se under heat stress condition, leading to an increase in live body weight and then a decrease in the dressed carcass weight [36].

One of the strongest possible explanations for the improved quail performance with nano-Se supplementation might be the antioxidant activity of Se. Our study reveals that the augmented oxidative stress that was induced under the effect of heat stress, represented by decreased blood GPx, SOD activities, and GSH level, along with an increased MDA level and the oxidative status of quails, was recovered under the effect of 0.2 mg/kg nano-Se supplementation. These data support similar outcomes of several previous reports [37,38,39]. The reason behind the oxidative booster effect of Se is its incorporation into selenocysteine (SeCys), which then forms selenoenzymes such as glutathione peroxidase [4]. Moreover, a previous investigation demonstrated that Se nanoformulations that are 100 to 500 nm provoked a higher antioxidative effect with an increased free radical entrapment capacity and a higher adsorptive ability compared to organic and inorganic forms due to enhanced interactions between NPs and proteins, structural functional groups, COO-, and C-NH [40].

However, Se is being used as a supplement for humans and livestock, Se’s potential benefits are not without risk, as there is a narrow window for doses that result in efficacy and doses that could be more harmful [7]. Our data reveal an oxidative adverse effect after a high dose of nano-Se supplementation (0.5 mg/kg), represented by decreased blood GPx, SOD activities, and GSH level along with an increased MDA level. Oxidative stress was previously reported in lab animals supplemented with different doses and sizes of Se and nano-Se [41,42,43]. Our explanation for the increase in oxidative stress with nano-Se supplementation under 0.5 mg/kg, in groups 3 and 6, is based on previous investigations that concluded this was due to the pro-oxidant Se property at relatively high doses and prolonged administration. Elemental Se can be converted by the cells to H_2_Se and selenols, which are readily oxidized by O_2_, which in turn produce ROS and consume intracellular antioxidants, such as GSH [44,45,46,47]. Selenols increase the production of superoxide radicals and react with thiols and di selenides to produce selenyl-sulphides and disulphides. This can cause protein aggregation, transcription factor inactivation, and disturbance in redox- regulated cell signaling [7]. Moreover, margeret et al. (2019) proposed that nano-Se has a hermetic effect, which describes a biphasic dose-dependent response of an agent with beneficial effects at low doses and adverse effects at high doses [8].

Our results revealed that heat stress and/or 0.2 and 0.5 nano-Se in combination with ambient temperature in quail groups affected the expression profile of inflammatory and antioxidant markers. The heat stress-exposed quails’ expression patterns of *IL-2, IL-4, IL-6*, and *IL-8* demonstrated a significant up-regulation as compared to the control group. Compared to the heat stress group, there was a significant decrease in the levels of the genes encoding *IL-2, IL-4, IL-6*, and *IL-8* and an increase in the levels of *SOD* and *GPX*, indicating the mitigating effect of 0.2 nano-Se. When quails exposed to 0.5 nano-Se are compared to the control group, their expression patterns of *IL-2, IL-4, IL-6*, and *IL-8* are significantly higher. Conversely, there was a noticeable downregulation of *GPX* and *SOD* markers. The expression profile of the genes under investigation was significantly impacted by the interaction between environmental temperature and various nano-Se concentrations (*p* < 0.01). To our knowledge, there is limited information on the effect of nano-se on gene expression profile of inflammatory and antioxidant markers in heat stress exposed quails. However, according to Calik et al. (2022), heat stress markedly decreased the levels of *TLR2*, *TNFa*, and increased those of *HSP70, HSP90, IL-4, IL-22, IFNg*, and *ZO*-1. On the other hand, mRNA abundance of *TLR2, TNFa, IFNg, IL-1b, IL-10*, and *iNOS* was considerably down-regulated in the liver of birds fed on Vit E/Se supplemented diet [8]. According to Li et al. (2021) study, nano-Se has an antagonistic effect on hepatocyte apoptosis caused by Di-(2-ethylhexyl)-phthalate (DEHP) via reducing the expression of apoptotic markers [48]. Other research study revealed that dietary Se, vitamin E, and Se+ vitamin E significantly increased the activity and mRNA levels of *CAT* and *SOD* but lowered the mRNA levels of HSP70 and HSP90 [36].

Because of its ability to lower gut immunity and the body’s antioxidant capacity, heat stress is a key cause of systemic oxidative stress [49]. According to Hedger and Meinhardt (2003), cytokines are small regulating proteins involved in the formation of immune cells, inflammation, and immunological responses [50]. According to Quinteiro-Filho et al. (2010) and Siddiqui et al. (2020), hyperthermia modifies cytokine expression and the immunological response [51,52]. During oxidative stress, dietary vitamin E and selenium can work in concert to affect immunological and antioxidant responses [53]. Thus, the cytokine modulation seen in this study is related to Se’s antioxidant properties. This size of the nanoparticles exhibits high levels of antioxidant activity [54], a greater capacity to trap free radicals with an antioxidant effect [55], and a greater capacity to absorb substances due to interactions with protein functional groups (NH, C=O COO-, and C-N) [39]. Furthermore, when given at a smaller particle size, nano-Se can function as a chemopreventive agent [56]. When combined, the study’s results offer strong proof that dietary nano-Se increases the expressions of SOD and CAT mRNAs in the liver, which in turn reduce the amount of MDA (lipid peroxidation) in the livers of quails exposed to high temperatures [57].

In this context, the *IL-2, IL-4, IL-6*, and *IL-8* genes were significantly upregulated in 0.5 nano-Se-, heat-stress-, and heat stress with 0.5 nano-Se-exposed quails. However, SOD and GPX markers showed an opposite trend. The elevation in cytokines, as well as decrease in antioxidant genes’ expression, could be attributed to the adverse effect of selenium. When too many ROS are present in cells and interact with other cellular constituents, genotoxicity results. By reacting with both deoxyribose sugars and DNA nucleobases, this results in base lesions and breaks apart deoxyribose nucleic acid (DNA) strands. Furthermore, ROS and Se may oxidize DNA and disrupt transcriptional control and DNA repair, endangering the integrity of genetic information. Nano-Se toxicity appears to be influenced by several interconnected factors that affect the organism’s biological response, including dose, exposure duration, and the size and chemistry of the nano-Se nanoparticles. According to Bano et al. (2022), the main targets of nano-Se toxicity are not only their pro-oxidative properties but also their interactions with metabolic and molecular signaling pathways, including apoptotic pathways; the ability of small nanoparticles to penetrate different tissues; and the organism’s ability to eliminate Se through enzymatic transformation. These findings are supported by the results of toxicological studies [58].

## 5. Conclusions

Varying levels of Nano-Selenium alongside environmental temperature could affect the growth performance, serum metabolites, and gene expression of heat-stressed growing quails. Supplementing the diets of heat-stressed growing quails with nano-Se especially 0.2 mg/kg of diet improved growth performance without any detrimental effects on carcass traits, blood parameters, or gene expression.

## Figures and Tables

**Figure 1 vetsci-11-00228-f001:**
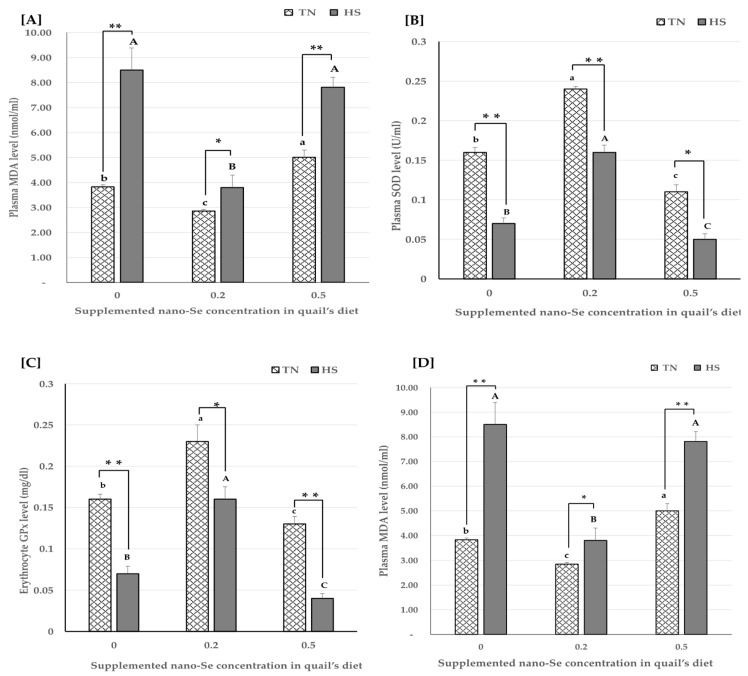
The effect of different nano-Se levels and environmental temperature on quail’s blood lipid peroxidation and antioxidant enzymes levels. MDA (**A**), SOD (**B**), GPx (**C**) and GSH (**D**) levels were measured in blood samples and data are presented as mean ± SEM. Different upper-case letters (A–C) show significant differences (*p* < 0.05) among different concentrations of nano-Se supplementation under heat stress condition. Different lower-case letters (a–c) exhibit significant differences (*p* < 0.05) among different concentrations of nano-Se supplementation under thermoneutral condition. Asterisk (*) represents statistical difference between thermoneutral and heat stress subjected groups, * *p* < 0.05, ** *p* < 0.01.

**Figure 2 vetsci-11-00228-f002:**
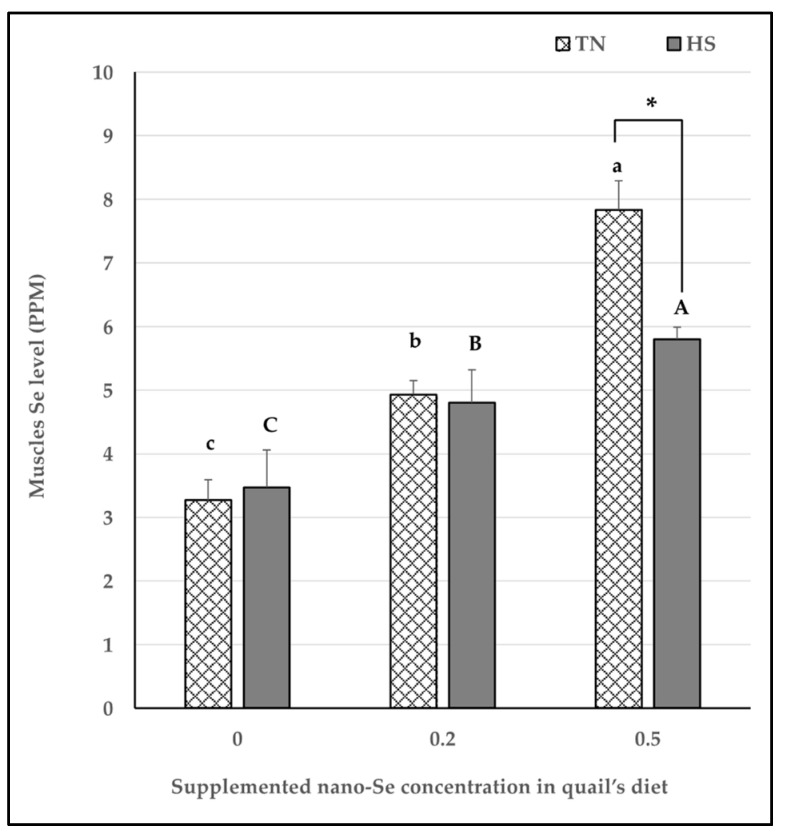
Muscles Se levels under the effect of different nano-Se supplementations and different environmental temperature. Se was measured in muscle tissue samples by ICP-MS. Data are presented as mean ± SEM. Different upper-case letters (A–C) show significant differences (*p* < 0.05) among different concentrations of nano-Se supplementation under heat stress condition. Different lower-case letters (a–c) exhibit significant differences (*p* < 0.05) among different concentrations of nano-Se supplementation under thermoneutral condition. Asterisk (*) represents statistical difference between thermoneutral and heat stress subjected groups, * *p* < 0.05.

**Figure 3 vetsci-11-00228-f003:**
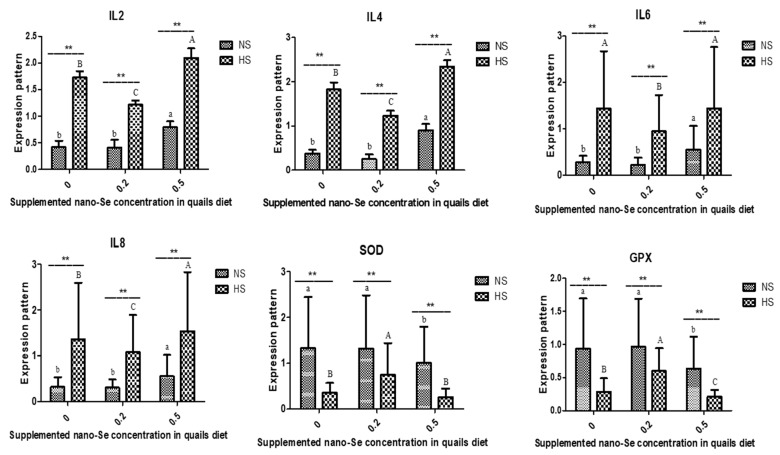
Effect of different nano-Se levels and environmental temperature on gene expression of inflammatory and antioxidant markers. Data are presented as mean ± SEM. Different upper-case letters (A–C) show significant differences (*p* < 0.05) among different concentrations of nano-Se supplementation under heat stress condition. Different lower-case letters (a–c) exhibit significant differences (*p* < 0.05) among different concentrations of nano-Se supplementation under thermoneutral condition. Asterisk (**) represents statistical difference between thermoneutral and heat stress subjected groups, ** *p* < 0.01.

**Table 1 vetsci-11-00228-t001:** Ingredients and proximate composition of the experimental diets.

Ingredients	%
Yellow corn	57.1
Soybean meal	30
Corn gluten	2
Wheat bran	7.3
Soyabean oil	0.5
Limestone	1.5
Dicalcium p	0.7
Premix *	0.3
Salt	0.3
DL methionine	0.1
Lysine DL	0.2
Proximate calculated (%) **	
CP%	23.92
ME, kcal/kg	2901.6
Ca%	1.18
Available P% Analyzed se content (mg/kg)	0.7 2.32

* Supplies per kg diet: Vitamin A, 16,500 IU; vitamin D3, 750 IU; vitamin E, 12 IU; vitamin K, 2 mg; vitamin B1, 1.2 mg; vitamin B2 10mg; vitamin B6, 2.4 mg; vitamin B12, 12 µg; niacin, 18 mg; pantothenic acid, 12 mg; Mn, 190 mg as manganese sulfate; Zn, 72 mg as zinc oxide; Fe, 380 mg as ferrous sulfate; copper, 13 mg as copper sulfate; iodine, 0.4 mg as potassium iodide. ** Calculated according to NRC (1994) for quails [21].

**Table 2 vetsci-11-00228-t002:** Sequence, accession number, annealing temperature, and PCR product size of oligonucleotide primers used in real-time PCR for inflammatory and antioxidant genes.

Gene	Isolation Source	Primer	Product Length (bp)	Annealing Temperature (°C)	Accession Number
*IL-2*	Spleen	F: GTGCAAAGTACTGATCTTCGCC R: CTTGGTGTGTAGAGCTCGAGATG	195	60	AY613440.1
*IL-4*	Spleen	F: GAGAGCATCCGGATAGTGAAG R: TTCGCATAAGAGCTGGGTTC	168	62	AB559571
*IL-6*	Spleen	F: CAACCTCAACCTGCCCAA R: GGAGAGCTTCCTCAGGCATT′	85	60	AB5595724
*IL-8*	Spleen	F: CTGAGGTGCCAGTGCATTAG R: AGCACACCTCTCTTCCATCC	139	58	AB559573
*SOD*	Liver	F: TGGACCTCGTTTAGCTTGTG R: ACACGGAAGAGCAAGTACAGR	126	62	NM_205064.1
*GPX*	Liver	F: TTGTAAACATCAGGGGCAAA R: TGGGCCAAGATCTTTCTGTAA	140	58	NM_001163245.1
*β-Actin*		F: CTGGCACCTAGCACAATGAA R: CTGCTTGCTGATCCACATCT	123	55	AF199488

Abbreviations: *IL-2* = interleukin-2; *IL-4* = interleukin-4; *IL-6* = interleukin-6; *IL-8* = interleukin-8; *SOD* = superoxide dismutase; *GPX* = glutathione peroxidase; *β-Actin =* beta actin.

**Table 3 vetsci-11-00228-t003:** Assessing the impacts of various amounts of nano-Se on the FBW, BWG, and FCR of heat-stressed quails over the experimental period.

	Environmental Temperature	TN			HS			Two Way Anova Analysis		
	Con. of Se (mg/kg)	0	0.2	0.5	0	0.2	0.5	Temp	Conc	Inter-action
**IBW (g)**		68.33 ± 2.07	68.33 ± 1.66	69.16 ± 2.87	70.4 ± 2.42	70.83 ± 3.18	70 ± 2.46	ns	ns	ns
**FBW (g)**		205.00 ^b^ ± 4.9	207.92 ^b^ ± 3.8	205.83 ^b^ ± 6.3	203.75 ^b^ ± 3.89	230.42 ^a^ ± 7.5	217.9 ^ab^ ± 4.5	**	*	*
**BWG(g)**		136.7 ^b^ ± 3.09	139.5 ^b^ ± 2.4	136.7 ^b^ ± 3.5	133.3 ^c^ ± 2.07	159.58 ^a^ ± 4.6	147.9 ^ab^ ± 2.4	**	**	**
**FCR**		4.57 ^a^ ± 0.12	4.08 ^bc^ ± 0.08	4.13 ^bc^ ±0.09	4.37 ^ab^ ± 0.24	3.79 ^c^ ± 0.11	4.07 ^bc^ ±0.06	**	**	ns

Data are presented as the mean ± SEM. Values with a different letter superscript in the same row indicate a significant difference between groups (*p* < 0.05). Two way a nova analysis was presented in the table, ns *p* > 0.05, * *p* < 0.05, ** *p* < 0.01.

**Table 4 vetsci-11-00228-t004:** The influence of different nano-Se concentrations and environmental temperature on carcass traits.

Parameters	Temperature	TN			HS			Tow Way Anova Analysis		
	Con. of Se (mg/kg)	0	0.2	0.5	0	0.2	0.5	Temp	Conc	Inter-action
**Dressed carcass weight**		133.3 ^ab^ ± 2.73	145 ^a^ ± 0.577	143.66 ^a^ ± 5.8	127.3 ^ab^ ± 5.2	136.3 ^ab^ ± 5.7	117.3 ^b^ ± 3.5	*	*	*
**Liver %**		1.96 ± 0.22	1.69 ± 0.34	2.5 ± 0.17	2.02 ± 0.16	2.5 ± 0.13	2.03 ± 0.03	ns	ns	ns
**Heart %**		0.968 ± 0.07	0.875 ± 0.01	0.914 ± 0.08	0.878 ± 0.02	0.743 ± 0.02	0.859 ± 0.04	ns	ns	ns
**Spleen**		0.115 ± 0.03	0.073 ± 0.02	0.123 ± 0.03	0.200 ± 0.05	0.163 ± 0.04	0.175 ± 0.03	ns	ns	ns

Data are presented as the mean ± SEM. Values with a different letter superscript in the same row indicate a significant difference between groups (*p* < 0.05). Tow way a nova analysis was presented in the table, ns *p* > 0.05, * *p* < 0.05.

## Data Availability

Upon justifiable demand, the supportive data for the findings of the study will be provided by the relevant author.

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
