# Peer review of "Assessing the Impacts of Different Levels of Nano-Selenium on Growth Performance, Serum Metabolites, and Gene Expression in Heat-Stressed Growing Quails"

_vetsci, 2024, doi:10.3390/vetsci11060228_

Round 1

Reviewer 1 Report

Comments and Suggestions for Authors

The authors analyzed the effects of 2 levels of nano-Se on growing Japanese quail. Growth performance until 40 days, plasma, viscera, and carcass were analyzed to antioxidant enzymes and lipid peroxidation. Additionally, genes for interleukins, SOD, and GPx were analyzed in liver and spleen.

The subject is adequate with the journal's scope. The argument to support the research is valid, the manuscript is well written and understandable to a specialist readership, and organization, and the article's structure is good and in agreement with the journal instructions for authors. However, I don't agree with the statistical analysis. In this experimental design, it is clear a factorial 2x3 with 2 environmental conditions and 3 levels of nano-Se in the diet. Authors need to test the isolated effect of each source of variation and their interaction. This is necessary to affirm, for example, that heat stress negatively affected the growth and feed utilization of quails and increased oxidative stress (line 43). Another important fact is the heat stress conditions are not cited in the methodology. This is one of the main sources of variation in data and needs to be very well described. These 2 points are my main considerations.

The discussion is relevant and well-written in the present format but can be very different, mainly based on the results, after the factorial analysis. For example, the Tukey differences only make sense if there is an interaction between environmental temperature and nano-Se level.

 I recommend the acceptance of the paper after Major Revision considering these comments, especially because a factorial analysis will change the results sections.

General comments – check the journal’s format to references list (a lot of mistakes); the format of authors especially commas after et al. and previously the year in parenthesis. Ex: Saurai et al., (2006) or Saurai et al. (2006)?

Also, when an author is cited in the text the number of references is inside [number] because a lot of them are missed.

In the results and discussion check along the text where authors are writing about genes use the italic format to their names. In some places it is adequate but a lot of missing text.

Specific comments are described below.

1)     Title – Suggestion: delete “ Assessing the impact of different levels of “

2)     Abstract – the experimental design needs to be changed to a factorial according to the suggestion in the statistical section. Even if the authors disagree with this suggestion, make it simple and direct. Insert data about nano-Se and the heat stress condition.

Suggestion, change these lines:

Group 1 was provided with a basal diet devoid of additional selenium, serving as the control group. Groups 2 and 3 were given a basal diet supplemented with 0.2 and 0.5 mg/kg of diet from nano-Se, respectively. Group 4 received a basal diet and was exposed to heat stress. Lastly, groups 5 and 6 were provided with a basal diet enriched with 0.2 and 0.5 mg/kg of diet from nano-Se, respectively, and subjected to heat stress.

To:  A randomized experimental design was used in a 2 × 3 factorial, with 2 environmental conditions (normal and heat stress) and 3 nano-Se levels (0, 0.2, and 0.5 mg/kg of diet). Nano-Se was provided as …………….. The heat stress consisted in ………………

After the factorial analysis, a different form to show the results will be necessary, and the abstract and results section needs to be modified accordingly.

3)     Key words – avoid words from the title. 

4)     Introduction – please insert the hypothesis before the objective.

5)     Methodology

Most important, there is no information about heat stress conditions. How long? What temperature? How was it obtained?

Item 2.6.1 is written in the future “will be” change the verb tense to past.

Line 192 correct Se.

Table 5 has results of MDA, GPx, GSH, SOD here SOD protocol is missing.

Table 1: describe the Premix composition, mixed oil

Correct - L-Lysine HCl …..%, DL-Methionine …..%, Dicalcium phosphate, Analyzed Se content (mg/kg)

Check the information the somatory of ingredients is 99.7 missed 0.3%

Statistics – comments in the first part.

Results

In the actual format, all tables show the means of interaction effects.

The tables are non-repetitive and concise, but many suggestions are below.

 All tables – consider deleting the first line; use nano-Se

Table 2: what is the **, ***, ****?

Line 270 - .2 or 0.2?

Table 4 – title: Table 4. The influence of varied nano-se levels in the diets of heat-stressed growing quails on carcass traits.

It is not adequate, first because the authors showed here data from normal temperatures too and it was at a specific age, then consider changing to an informative title.

 Table 5 – insert in the title: lipid peroxidation, and antioxidant enzyme concentration in the plasma and Se concentration in the muscle (?). there are two MDA specified inside the table. Perhaps a line with blood and another with Breast muscle separating data.

  The discussion is adequate.

 Conclusion – The conclusion section is a discussion. Move those lines or part of them to the right section.

Author Response

Comments and Suggestions for Authors

 Comment

The authors analyzed the effects of 2 levels of nano-Se on growing Japanese quail. Growth performance until 40 days, plasma, viscera, and carcass were analyzed to antioxidant enzymes and lipid peroxidation. Additionally, genes for interleukins, SOD, and GPx were analyzed in liver and spleen.

The subject is adequate with the journal's scope. The argument to support the research is valid, the manuscript is well written and understandable to a specialist readership, and organization, and the article's structure is good and in agreement with the journal instructions for authors. However, I don't agree with the statistical analysis. In this experimental design, it is clear a factorial 2x3 with 2 environmental conditions and 3 levels of nano-Se in the diet. Authors need to test the isolated effect of each source of variation and their interaction. This is necessary to affirm, for example, that heat stress negatively affected the growth and feed utilization of quails and increased oxidative stress (line 43). Another important fact is the heat stress conditions are not cited in the methodology. This is one of the main sources of variation in data and needs to be very well described. These 2 points are my main considerations.

Response

We are grateful to the reviewer for drawing it to our consideration. The statistical analysis is changed according to inquiry.

The discussion is relevant and well-written in the present format but can be very different, mainly based on the results, after the factorial analysis. For example, the Tukey differences only make sense if there is an interaction between environmental temperature and nano-Se level.

 I recommend the acceptance of the paper after Major Revision considering these comments, especially because a factorial analysis will change the results sections.

Comment

General comments – check the journal’s format to references list (a lot of mistakes); the format of authors especially commas after et al. and previously the year in parenthesis. Ex: Saurai et al., (2006) or Saurai et al. (2006)?

Response

We are grateful to the reviewer for drawing it to our consideration. The references are revised.

Comment

Also, when an author is cited in the text the number of references is inside [number] because a lot of them are missed.

Response

We are grateful to the reviewer for drawing it to our consideration. The references are revised.

Comment

In the results and discussion check along the text where authors are writing about genes use the italic format to their names. In some places it is adequate but a lot of missing text.

Response

We are grateful to the reviewer for drawing it to our consideration. Done

Specific comments are described below.

Coment

1)     Title – Suggestion: delete “ Assessing the impact of different levels of “

 Response

We are grateful to the reviewer for drawing it to our consideration. Done

Comment

2)     Abstract – the experimental design needs to be changed to a factorial according to the suggestion in the statistical section. Even if the authors disagree with this suggestion, make it simple and direct. Insert data about nano-Se and the heat stress condition.

 Response

We are grateful to the reviewer for drawing it to our consideration. Done

Suggestion, change these lines:

Comment

Group 1 was provided with a basal diet devoid of additional selenium, serving as the control group. Groups 2 and 3 were given a basal diet supplemented with 0.2 and 0.5 mg/kg of diet from nano-Se, respectively. Group 4 received a basal diet and was exposed to heat stress. Lastly, groups 5 and 6 were provided with a basal diet enriched with 0.2 and 0.5 mg/kg of diet from nano-Se, respectively, and subjected to heat stress.

To:  A randomized experimental design was used in a 2 × 3 factorial, with 2 environmental conditions (normal and heat stress) and 3 nano-Se levels (0, 0.2, and 0.5 mg/kg of diet). Nano-Se was provided as …………….. The heat stress consisted in ………………

After the factorial analysis, a different form to show the results will be necessary, and the abstract and results section needs to be modified accordingly.

Response

We are grateful to the reviewer for drawing it to our consideration. Done

Comment

3)     Key words – avoid words from the title. 

Response

We are grateful to the reviewer for drawing it to our consideration. Done

Comment

4)     Introduction – please insert the hypothesis before the objective.

Response

We are grateful to the reviewer for drawing it to our consideration. Done.

5)     Methodology

Comment

Most important, there is no information about heat stress conditions. How long? What temperature? How was it obtained?

Response

We thank the reviewer for this. Done.

Comment

Item 2.6.1 is written in the future “will be” change the verb tense to past.

Response

We thank the reviewer for this. Corrected.

Comment

Line 192 correct Se.

Response

We thank the reviewer for this. Corrected.

Comment

Table 5 has results of MDA, GPx, GSH, SOD here SOD protocol is missing.

Response

We thank the reviewer for this. The results are changed into figures for better presentation of data.

Response

We thank the reviewer for this. Corrected.

Comment

Table 1: describe the Premix composition, mixed oil

Correct - L-Lysine HCl …..%, DL-Methionine …..%, Dicalcium phosphate, Analyzed Se content (mg/kg)

Check the information the somatory of ingredients is 99.7 missed 0.3%

 Response

We are grateful to the reviewer for drawing it to our consideration. Done

Statistics – comments in the first part.

Comment

Results

In the actual format, all tables show the means of interaction effects.

The tables are non-repetitive and concise, but many suggestions are below.

 All tables – consider deleting the first line; use nano-Se

Table 2: what is the **, ***, ****?

  Response

We are grateful to the reviewer for drawing it to our consideration. Done

Comment

Line 270 - .2 or 0.2?

Response

We are grateful to the reviewer for drawing it to our consideration. Corrected

Comment

Table 4 – title: Table 4. The influence of varied nano-se levels in the diets of heat-stressed growing quails on carcass traits.

It is not adequate, first because the authors showed here data from normal temperatures too and it was at a specific age, then consider changing to an informative title.

Response

We are grateful to the reviewer for drawing it to our consideration. Corrected

 Comment

Table 5 – insert in the title: lipid peroxidation, and antioxidant enzyme concentration in the plasma and Se concentration in the muscle (?). there are two MDA specified inside the table. Perhaps a line with blood and another with Breast muscle separating data.

Response

We thank the reviewer for this. The results are changed into figures for better presentation of data.

  Comment

The discussion is adequate.

Response

We thank the reviewer for this positive comment.

Comment

 Conclusion – The conclusion section is a discussion. Move those lines or part of them to the right section.

Response

We thank the reviewer for this. Corrected.

Reviewer 2 Report

Comments and Suggestions for Authors

In the interesting manuscript Rania Mahmoud et al. investigated the effects of different concentrations of nanosized selenium on the growth performance, carcass features, serum metabolites and gene expression of heat stressed quails. The experiments are well designed. However, the quality of articles and tables needs to be strongly improved. With editing and some revisions, I feel that this manuscript will be suitable for publication.

1. Line 153-155: What is the purpose of selenium in the diet as determined by atomic absorption spectrophotometry? How do the results of the determination relate to the amount of nano-Se added?

2. Table 1: "Available P,%"and "Analyzed se content(mg/kg)" have been incorrectly moved down a row, please redraw the table.

3. Section 2.4: Why were 16 quails selected for weighing in each replicate?

4. Line 181: Replace "-20 C" with "-20".

5. Line 206: "5x" is a spelling mistake. There are many identical errors in the article. Please check it.

6. Table 2: Replace "Annealing temperature (C0)" with "Annealing temperature (℃)".

7. Table 34: The contents of the two tables are very confusing. Please remake neat tables.

8. Section 3.3 and Table 5: Please standardize the correct expression of numbers in the article. Please also pay attention to the correct form of writing the units.

9. Line 290: The names of the genes should be in italics, please correct it.

10. What is the ambient temperature of the heat stress group setup?

11. Why IL-2, IL-4, IL-6, and IL-8 are tested?

12. In addition to using tables, the article can also draw histograms to more intuitively display the experimental results.

13. P in P<0.05 should be in italics, please check the entire article for consistent formatting.

Author Response

Reviewer 2

Comments and Suggestions for Authors

In the interesting manuscript Rania Mahmoud et al. investigated the effects of different concentrations of nanosized selenium on the growth performance, carcass features, serum metabolites and gene expression of heat stressed quails. The experiments are well designed. However, the quality of articles and tables needs to be strongly improved. With editing and some revisions, I feel that this manuscript will be suitable for publication.

  1. Line 153-155: What is the purpose of selenium in the diet as determined by atomic absorption spectrophotometry? How do the results of the determination relate to the amount of nano-Se added?

Response

We are grateful to the reviewer for drawing it to our consideration. Clarified.

Comment

  1. Table 1: "Available P,%"and "Analyzed se content(mg/kg)" have been incorrectly moved down a row, please redraw the table.

Response

We are grateful to the reviewer for drawing it to our consideration. Corrected

Comment

  1. Section 2.4: Why were 16 quails selected for weighing in each replicate?

Response

We thank the reviewer for this. We have randomly selected 16 quail in each replicate as representative samples as a trial without any bias. We have relied on random method in getting quails. We think this number could be adequate in judgment.

Previous literatures made assessments based on these numbers or less than less. Any how we have dded the word randomly to the manuscript..

Comment

  1. Line 181: Replace "-20 C" with "-20℃".

Response

We are grateful to the reviewer for drawing it to our consideration. Corrected

Comment

  1. Line 206: "5x" is a spelling mistake. There are many identical errors in the article. Please check it.

Response

We are grateful to the reviewer for drawing it to our consideration. Corrected

Comment

  1. Table 2: Replace "Annealing temperature (C0)" with "Annealing temperature (℃)".

Response

We are grateful to the reviewer for drawing it to our consideration. Corrected

Comment

  1. Table 3、4: The contents of the two tables are very confusing. Please remake neat tables.

Response

We are grateful to the reviewer for drawing it to our consideration. Corrected

Comment

  1. Section 3.3 and Table 5: Please standardize the correct expression of numbers in the article. Please also pay attention to the correct form of writing the units.

Response

We are grateful to the reviewer for drawing it to our consideration. Corrected

Comment

  1. Line 290: The names of the genes should be initalics, please correct it.

Response

We are grateful to the reviewer for drawing it to our consideration. Corrected

Comment

  1. What is the ambient temperature of the heat stress group setup?

Response

We are grateful to the reviewer for drawing it to our consideration. Clarified.

Comment

  1. Why IL-2, IL-4, IL-6, and IL-8 are tested?

Response

We are grateful to the reviewer for this. Heat stress significantly impacts the immunity and cytokine expression of poultry. In addition to the above, cytokines play an important role in healing injured tissue, including insults arising as a result of heat stroke. We tried to investigate the effect of heat stress on cytokine genes expression and the possible potential ameliorative effect of nanoselenium.

Body temperature is regulated by cytokines, which are defined as regulatory proteins of polypeptides produced by immune cells in response to tissue injury, infection, stress, or inflammation

Saleh KMM, Al-Zghoul MB. Effect of Acute Heat Stress on the mRNA Levels of Cytokines in Broiler Chickens Subjected to Embryonic Thermal Manipulation. Animals (Basel). 2019 Jul 29;9(8):499. doi: 10.3390/ani9080499. PMID: 31362400; PMCID: PMC6719976.

Comment

  1. In addition to using tables, the article can also draw histograms to more intuitively display the experimental results.

Response

We are grateful to the reviewer for drawing it to our consideration. Done

Comment

  1. P in P<0.05 should be in italics, please check the entire article for consistent formatting.

 Response

We are grateful to the reviewer for drawing it to our consideration. Done

Round 2

Reviewer 2 Report

Comments and Suggestions for Authors

Thank you very much!